# Small Heat Shock Protein 22 Improves Cognition and Learning in the Tauopathic Brain

**DOI:** 10.3390/ijms23020851

**Published:** 2022-01-13

**Authors:** Santiago Rodriguez Ospina, Danielle M. Blazier, Marangelie Criado-Marrero, Lauren A. Gould, Niat T. Gebru, David Beaulieu-Abdelahad, Xinming Wang, Elizabeth Remily-Wood, Dale Chaput, Stanley Stevens, Vladimir N. Uversky, Paula C. Bickford, Chad A. Dickey, Laura J. Blair

**Affiliations:** 1USF Health Byrd Alzheimer’s Institute, University of South Florida, Tampa, FL 33613, USA; santiago3@usf.edu (S.R.O.); dblazier@usf.edu (D.M.B.); marangelie@usf.edu (M.C.-M.); gouldl@acom.edu (L.A.G.); niat@usf.edu (N.T.G.); davidbeaulie@usf.edu (D.B.-A.); xxw806@case.edu (X.W.); vuversky@usf.edu (V.N.U.); 2Department of Molecular Medicine, University of South Florida, Tampa, FL 33612, USA; erwood@usf.edu; 3Department of Molecular Pharmacology and Physiology, University of South Florida, Tampa, FL 33612, USA; pbickfor@usf.edu; 4Department of Cell Biology, Microbiology and Molecular Biology, University of South Florida, Tampa, FL 33620, USA; chaput@usf.edu (D.C.); smstevens@usf.edu (S.S.Jr.); 5Center for Molecular Mechanisms of Aging and Age-Related Diseases, Moscow Institute of Physics and Technology, Institutskiy Pereulok, 9, 141700 Dolgoprudny, Russia; 6Research Service, James A. Haley Veterans’ Hospital, Tampa, FL 33620, USA; 7Department of Neurosurgery and Brain Repair, University of South Florida, Tampa, FL 33613, USA

**Keywords:** tau, synaptic plasticity, Alzheimer’s disease, molecular chaperone, proteomics, mass spectrometry

## Abstract

The microtubule-associated protein tau pathologically accumulates and aggregates in Alzheimer’s disease (AD) and other tauopathies, leading to cognitive dysfunction and neuronal loss. Molecular chaperones, like small heat-shock proteins (sHsps), can help deter the accumulation of misfolded proteins, such as tau. Here, we tested the hypothesis that the overexpression of wild-type Hsp22 (wtHsp22) and its phosphomimetic (S24,57D) Hsp22 mutant (mtHsp22) could slow tau accumulation and preserve memory in a murine model of tauopathy, rTg4510. Our results show that Hsp22 protected against deficits in synaptic plasticity and cognition in the tauopathic brain. However, we did not detect a significant change in tau phosphorylation or levels in these mice. This led us to hypothesize that the functional benefit was realized through the restoration of dysfunctional pathways in hippocampi of tau transgenic mice since no significant benefit was measured in non-transgenic mice expressing wtHsp22 or mtHsp22. To identify these pathways, we performed mass spectrometry of tissue lysates from the injection site. Overall, our data reveal that Hsp22 overexpression in neurons promotes synaptic plasticity by regulating canonical pathways and upstream regulators that have been characterized as potential AD markers and synaptogenesis regulators, like EIF4E and NFKBIA.

## 1. Introduction

Tau is a microtubule-associated protein that has been linked to multiple neurodegenerative diseases, with the most prevalent being Alzheimer’s disease (AD) [1]. In AD and other tauopathies, tau pathologically aggregates, which has been identified as a main contributing factor to the pathogenicity of the disease [2]. Currently, there are no effective therapies to treat patients with tauopathies [3], so the identification of additional molecular targets that can regulate tau aggregation and toxicity may provide novel therapeutic strategies.

Molecular chaperones, including heat-shock proteins (Hsps), are essential regulators of cellular homeostasis that assist in protein folding, refolding, and degradation [4,5]. Small Hsps (sHsps) are highly conserved, ATP-independent molecular chaperones [6,7,8]. Instead of ATP, the activity of sHsps is regulated through phosphorylation of their N-terminal domain through stress-activated kinases [9], which can be simulated by phosphomimetic substitutions of corresponding amino acids [10]. These sHsps have been shown to have increased expression with aging, to directly interact with autophagy-related proteins, and to form complexes that help regulate aberrant proteins in the cell [11,12,13].

Discrete sHsps have also been described as potent inhibitors of protein aggregation [8]. Hsp27, also known as HSPB1, has been shown to inhibit aggregation of tau and other aggregating proteins, such as Aβ and α-synuclein [14,15,16,17]. We previously showed that Hsp27 slows tau accumulation in vitro and in vivo [18]. In this work, we found that the cycling of Hsp27 phosphorylation was essential for controlling tau accumulation and promoting long-term potentiation (LTP) in vivo. The closely related chaperone Hsp22, encoded by the *HSPB8* gene, has been shown to reduce Aβ and α-synuclein aggregation and has a protective role during aging [16,19,20]. Unlike Hsp27, which forms large homo-oligomeric structures that are disrupted by phosphorylation, Hsp22 forms dimers, which are stabilized by phosphorylation at S24 and/or S57 [21,22]. In fact, Hsp22 phosphorylation or mimetic mutations can impair the chaperone activity of Hsp22, as previously demonstrated by reduced inhibition of insulin and rhodanese aggregation [21]. Our recent work revealed that Hsp22 also regulates tau aggregation and tau phase separation in vitro [12,23], which was not affected by mutations that mimic or block phosphorylation. However, the impact of Hsp22 on tau in vivo and how this is influenced by phosphorylation has not been explored. Here, we investigated the effects of overexpressing wild-type and phosphomimetic mutant Hsp22 on tau accumulation, cognitive function, synaptic plasticity, and neuronal health in the brains of tau transgenic mice.

## 2. Results

### 2.1. mtHsp22 Preserves Spatial Reversal Learning and Memory in rTg4510 Mice

To understand the effects of Hsp22 overexpression on tau in the brain, 4-month-old rTg4510 tau transgenic and non-transgenic mice were bilaterally injected with adeno-associated virus (AAV) serotype 9-expressing wild-type Hsp22 (wtHsp22), S24,57D phosphomimetic mutant Hsp22 (mtHsp22), or GFP control (Figure 1a). All animals used are summarized in Table 1. After 2 months of viral expression, Hsp22 levels were evaluated, which revealed a significant upregulation in Hsp22 injected mice compared to GFP controls by immunostaining (Figure 1b–e) and quantitative mass spectrometry (MS) parallel reaction monitoring (PRM) (Figure 1f,g). Data indicate there is more expression for mtHsp22 in comparison with wtHsp22 for both non-transgenic and rTg4510 mice. Directly before tissue processing, mice were tested for spatial learning and memory by two-day radial-arm water maze (RAWM) with reversal. This task was selected because rTg4510 mice show impairments [24,25], which may be relevant to cognitive deficits found in AD patients [26]. mtHsp22 overexpression improved spatial reversal learning in rTg4510 mice (Figure 2a) and trended towards significance in mice injected with wtHsp22, which was driven by the male mice (Appendix A). Non-transgenic mice were indistinguishable from one another, regardless of treatment, suggesting the Hsp22-mediated improvement in rTg4510 mice was not a general procognitive effect but a protection from tau-mediated deficits. It was previously demonstrated that rTg4510 mice show increased activity in the open-field task [27]. Overexpression of wtHsp22 and mtHsp22 protected rTg4510 mice from this hyperactive phenotype (Figure 2b), similarly to what was previously demonstrated with early modifiers of tau activity [27]. We confirmed that these changes were not caused by altered anxiety-like behavior in these mice based on treatment or genotype (Figure 2c). These data indicate that Hsp22 overexpression protects from tau-mediated deficits in spatial reversal learning and locomotion.

### 2.2. wtHsp22 and mtHsp22 Overexpression Rescue LTP Deficits in rTg4510 Mice

Since our prior work identified protective effects in synaptic plasticity following Hsp27 overexpression in rTg4510 mice [18], we investigated whether Hsp22 overexpression provides similar functional benefits. Hippocampal ex vivo slices were prepared for electrophysiological analysis from a portion of the injected mice following behavioral testing. wtHsp22 overexpression significantly enhanced LTP in rTg4510 mice compared to GFP-injected controls (Figure 3a). Non-transgenic mice did not show any change in LTP by treatment (Figure 3b), suggesting that the rescue in the rTg4510 was due to a protection from a tau-mediated deficit. To determine whether this benefit was caused by a specific upregulation in presynaptic or postsynaptic signaling, we analyzed the ratio of the input stimulus to evoke output signal. The input/output (I/O) analysis showed that the fiber volley and the slope of the field excitatory postsynaptic potential (fEPSP) were not affected by treatment in either genotype (Figure 3c,d). This indicates that the effects of wtHsp22 and mtHsp22 in rTg4510 mice were not due to changes in basal synaptic transmission or presynaptic activity.

### 2.3. Overexpression of wtHsp22 and mtHsp22 Does Not Change Tau Levels or Phosphorylation Status

Next, we wanted to determine whether the behavioral and electrophysiological benefits of wtHsp22 and mtHsp22 overexpression in tau transgenic mice were due to reduced tau accumulation. Immunohistochemical and immunoblotting analyses revealed no change in total tau (Figure 4a,b and Appendix A) in pS262 (Figure 4c,d and Appendix A) and pT231 (Figure 4e,f and Appendix A) phospho-tau species in the hippocampi of the injected rTg4510 mice. Then, we measured the levels of Gallyas-silver-positive tau (Figure 4g,h) and T22-positive tau oligomers (Figure 4i,j and S2) in the hippocampi of the injected rTg4510 mice, which also showed no significant changes by Hsp22 overexpression.

### 2.4. Hsp22 Overexpression Preserves Neurons in rTg4510 Mice

Since the rTg4510 mice overexpressing Hsp22 were protected against tau-mediated cognitive and synaptic deficits through RAWM and LTP, we examined the impact of Hsp22 overexpression on neuronal health using unbiased stereology. We focused on the dentate gyrus because neurons are lost early in this region in the rTg4510 mouse model [28,29]. Stereological counts in the dentate gyrus revealed that wtHsp22 overexpression preserved significantly more neurons than GFP overexpression in rTg4510 mice. mtHsp22-expressing rTg4510 mice showed a non-significant positive trend of total neurons in this area (Figure 5a,b). Since Hsp22 has been previously linked to neurogenesis and autophagy [30,31,32], we investigated markers of these pathways. Using doublecortin (DCX) as a readout of neurogenesis, we observed that wtHsp22 and mtHsp22 overexpression caused a non-significant increase in this marker (Appendix A). In addition, autophagy, as measured by p62 levels, was unchanged by treatment (Appendix A). This suggests that the functional benefits of wtHsp22 and mtHsp22 overexpression may be realized through the modulation of other defective pathways in rTg4510 mice.

### 2.5. Overexpression of Hsp22 Changes Expression of Key Neuroprotective Proteins in rTg4510 Mice

For proteomic analysis, proteins were measured using MS of brain lysates from rTg4510 and non-transgenic mice injected with GFP (*n* = 6/genotype), wtHsp22 (*n* = 6 for rTg4510 and *n* = 7 for Non-Tg), and mtHsp22 (*n* = 6 for rTg4510 and *n* = 6 for Non-Tg) and quantified using LFQ (label-free quantification). The fold change of the differentially expressed proteins was used for further bioinformatic analyses to elucidate proteomic changes related to Hsp22 overexpression. We then plotted the expression-fold changes against the corresponding -log_10_(*p*-value) for each protein to generate volcano plots (Figure 6a–c). Validating the integrity of our data, MAPT (tau) was significantly upregulated in tau transgenic compared to non-transgenic mice, and HSPB8 (Hsp22) was significantly upregulated in wtHsp22- and mtHsp22-injected mice compared to GFP controls, further supporting our quantitative results from the MS PRM (Figure 1f,g).

### 2.6. Hsp22 Overexpression Protects against Tau-Mediated Disruption of Cellular Metabolism, Health, and Integrity Pathways

To understand how Hsp22 overexpression protected against tau-mediated defects, we used ingenuity pathway analysis (IPA) to explore relationships amongst the differentially expressed proteins. We focused on canonical pathways that were dysregulated in the GFP-injected rTg4510 compared to non-transgenic mice and restored by wtHsp22 or mtHsp22 overexpression. A comparison analysis was performed, and a heat map was created based on the trend of the z-scores between groups for the canonical signaling pathways, annotated using the core analysis module in IPA (Figure 7a,b). Since the functional benefits were strongest in the mtHsp22-overexpressing rTg4510 mice, we sorted the regulated pathways according to this z-score trend: pathways that were downregulated in GFP rTg4510 mice compared to GFP non-transgenic mice, semi-upregulated in wtHsp22 rTg4510 mice, and upregulated in mtHsp22 rTg4510 mice (Figure 7a). Another comparison analysis was performed, and a heat map was generated based on the opposite trends (Figure 7b). The pathways that were restored in rTg4510 mice by mtHsp22 overexpression and, to a lesser degree, by wtHsp22 overexpression were linked to neuronal protection, neuronal development, and cell growth (i.e., synaptogenesis signaling pathway and 14-3-3-mediated signaling pathway and mTOR signaling), as well as cellular energy (i.e., AMPK signaling) (Figure 7c–e). We also found that protein kinase A (PKA) was strongly downregulated by Hsp22 overexpression but was not altered by tau expression.

Next, upstream regulator analysis was performed to elucidate the proteins contributing to changes in the identified pathways. We sorted the proteins based on the trend of the z-scores for upstream regulators that were up- (Figure 8a) or downregulated (Figure 8b) in GFP-injected rTg4510 compared to GFP-injected non-transgenic mice, which were regulated by wtHsp22 or mtHsp22. The identified upstream regulators include transcription factors responsible for cellular integrity (i.e., STAT1, ERK1/2, JAK2) and other factors involved with neuronal and overall protein synthesis and cellular protection (i.e., EIF4E, NRAS, NFKBIA, LONP1, SIRT1). Due to the quantity of upstream regulators involved, we narrowed down and validated two hits relevant to AD and neuronal function: NFKBIA and EIF4E. Western blot analysis demonstrated that NFKBIA was downregulated in rTg4510 mice injected with wtHsp22 and mtHsp22 compared to GFP-injected mice (Figure 8c,d), and EIF4E was also decreased in mtHsp22 rTg4510 mice compared to GFP-injected mice (Figure 8c,e). We then performed a cluster analysis on these two upstream regulators in GFP rTg4510 and non-transgenic mice, wtHsp22 rTg4510 and GFP rTg4510 mice, and mtHsp22 rTg4510 and GFP rTg4510 mice (Appendix A). Surprisingly, these proteins contained noticeable levels of intrinsic disorder (26.7% and 38.5% of their sequences were predicted as disordered, respectively). Furthermore, we found that both proteins utilize portions of their intrinsically disordered regions for protein-protein interactions (they contain disorder-based protein-binding sites (Appendix A), molecular recognition features (MoRFs—residues 41–48 in EIF4E and residues 1–13, 76–83, and 299–314 in NFKBIA), or as sites of various post-translational modifications. It is likely that such prevalence of intrinsic disorder defines the multifunctionality of these proteins and their promising binding promiscuity, as illustrated by well-developed and dense protein-protein interaction networks centered around these hits. Overall, these data suggest that Hsp22 overexpression restored defective pathways in rTg4510 mice and regulated additional proteins important for learning and memory.

## 3. Discussion

In this study, we found that overexpressing Hsp22 in tau transgenic mice improved cognition and synaptic plasticity. This protection was not complemented by changes in tau but was concomitant with improved neuronal health, which was reflected in higher neuronal counts and the restoration of key neuroprotective proteins. Since no benefit was measured in non-transgenic mice, this indicates that Hsp22 overexpression protected against tau-mediated defects. Overall, this work suggests that high levels of Hsp22 may provide a safeguard against neurotoxic insults.

Our prior work demonstrated that the improved synaptic plasticity associated with Hsp27 overexpression in the tauopathic brain correlated with decreased tau accumulation [18]. Although there are differences in the readouts between these studies, beneficial effects of the Hsp27 mutant were not found in vivo [18]. In contrast, mtHsp22 enhanced LTP induction and maintenance in rTg4510 mice compared with wtHsp22. The enhanced benefits that mtHsp22 showed may be caused by the self-oligomerization differences between these chaperones. Hsp27 forms high-ended homo-oligomers that can be activated for chaperone activity by stress through phosphorylation, whereas Hsp22 has been shown to form dimers and other small complexes [22]. Herein, we utilized the S24/57D Hsp22 mutation, which was previously shown to affect the concentration-dependent association of Hsp22 subunits and chaperoning activity [21]. This phosphomimic maintained higher expression in the brain than its wild-type counterpart, suggesting this mutation—and possibly phosphorylation itself—may slow turnover of Hsp22 through self-stabilization [21]. Hsp22 has additional phosphorylation sites, as well as kinase activity (in fact, this chaperone is also known as protein kinase H11) [33], which have not yet been explored for their effect on tau aggregation or neuronal health. Recently, we reported that Hsp22 reduced tau levels in vitro, which was potentiated by an N-terminal deletion mutant (ΔNTD Hsp22) [12]. Additional studies are needed to test whether ΔNTD Hsp22 provides further benefits over mtHsp22 tested here since it is expected that ΔNTD Hsp22 may lower tau levels, in addition to the functional benefits we identified.

Here, we found Hsp22 overexpression protected rTg4510 mice from reversal spatial-learning deficits, particularly in male mice. This sex-dependent effect could be due to the worsened pathological and behavioral outcomes previously reported in female rTg4510 mice [34]. The functional benefits of Hsp22 overexpression in tau transgenic mice were found to be associated with the restoration of important cell-health and integrity pathways. It is possible that these pathways or complimentary pathways may have also contributed to the synaptic protection by Hsp27 in our prior study, but additional studies are needed to test this. However, Hsp27 has been shown to have protective roles in the brain after induced strokes by rendering resistance to excitotoxicity and preventing neuronal death by interfering with apoptosis [35,36]. This further suggests that sHsps have the capability to act through different pathways that can help maintain neuronal health. It is also possible that the protective effects in LTP and RAWM could be caused by a genotype effect or an indirect effect of tau that is being rescued by the overexpression of mtHsp22. In the case of a genotype effect, it is known that rTg4510 mice have gene disruptions caused by the insertion of the MAPT^P301L^ transgene, which cause an accelerated shrinking of the dentate gyrus, as well as cognitive deficits [37]. However, the rTg4510 model still recapitulates some features of the AD brain, like progressive tau accumulation, learning and memory deficits, and neuronal loss surpassing the number of neurofibrillary tangles [38], suggesting it is still a useful model, despite the caveats. Aberrant tau accumulation has also been shown to cause neuronal dysregulation by altering numerous pathways, which results in synaptic dysfunction, neuronal toxicity, and, ultimately, neuronal death [39]. Specifically, there is evidence that shows aberrant tau^P301L^ proteins are mislocalized to dendritic spines in rTg4510 mice, which impairs synaptic function through reduced glutamate receptors [40]. This suggests that mtHsp22 overexpression could be providing beneficial effects by affecting regulators of glutamate receptors, like AMPK signaling pathways, which were found to be upregulated by mtHsp22. While AMPK signaling has been associated with reduced hippocampal neurogenesis with aging [41], it has also been shown to be an essential regulator of synaptic plasticity by increasing postsynaptic receptors [42]. The canonical pathways identified to be regulated by Hsp22 in our study are all closely linked and may provide insight into the mechanisms underneath the functional benefits. AMPK acts downstream of PKA [43], which regulates glutamate receptor cycling [43], can inhibit mTOR signaling, which can regulate protein translation and synaptic plasticity [44,45], and is regulated by 14-3-3 proteins, which have key roles in regulating protein-protein interactions and modulate calcium influx [46,47]. Since each of these canonical pathways were found, in this study, to be altered by mtHsp22, it is possible that their regulation contributed to the protective effects of Hsp22 in the RAWM and LTP. Hsp22 has also been shown to promote macroautophagy with cochaperone BAG3, Hsp70, and p62 [48]. While we did not see a direct effect when staining for p62, suggesting that Hsp22 overexpression did not significantly upregulate autophagy, we cannot rule out that Hsp22 could have still promoted some protein clearance through this pathway, which has been associated with enhanced synaptic plasticity [49].

Evidence generated by MS analysis suggests that there was an activation of upstream regulator eukaryotic translation initiation factor 4E (EIF4E) in rTg4510 mice expressing wtHsp22 and mtHsp22 compared to GFP controls. This is interesting since EIF4E has been shown to be differentially expressed in AD patients [50]. Moreover, EIF4E phosphorylation is increased in the brains of AD patients and correlates with hyperphosphorylated tau [51]. EIF4E is an initiation factor that regulates ribosome interaction with the mRNA cap [52,53], controlling translation for a subset of proteins, like MMP9, PER1/2, and nuclear factor κ B inhibitor alpha (NFKBIA) [54]. EIF4E has been shown to correlate with NFKBIA levels [55]. In line with this, we found both proteins to be upregulated in rTg4510 mice and downregulated by Hsp22 overexpression. Literature has demonstrated, through genomic studies, that NFKBIA is upregulated in AD patients and could be used as a potential biomarker for AD [56,57]. NFKBIA is the primary negative regulator of NF-κB, an important transcription factor that has been suggested to have a role in synaptic plasticity by affecting LTP and neuronal health [58,59,60]. Our data further support this connection.

Our MS analysis also identified other differentially regulated proteins involved in neuronal health and synaptic plasticity, including NQO1, AKAP12, BAX, and PSMB10. NQO1 is an oxidoreductase enzyme that helps protect neurons against oxidative stress and has been shown to be upregulated by dopaminergic neuronal loss and in motor disorders [61]. AKAP12, is a member of a family of enzymes that help regulate PKA. A-kinase anchoring proteins (AKAP) have also been linked to maintaining cAMP compartmentalization, which is associated with learning and memory [62], and have been shown to interact with cadherin adhesion molecules at the postsynaptic density, which is important for synaptic plasticity [63]. BAX inhibition has been shown to protect neuronal cells from neurotoxicity [64]. PSMB10 is a subunit of the immunoproteasome, which is an important protein complex in the immune response that helps clear toxic proteins [65]. Literature has recently shown that the immunoproteasome is upregulated in AD, and it is believed that this sustained immune response can potentially cause inflammatory cell stress, leading to AD pathology [66]. Future studies are needed to explore the beneficial effects of targeting these proteins in tauopathy.

In conclusion, Hsp22 overexpression protected against tau-mediated deficits in neuronal health and cognition. Instead of significant changes in tau, we identified altered expression of proteins important for neuronal and overall cellular health by Hsp22 overexpression in rTg4510 mice. These proteins are part of canonical pathways related to synaptogenesis and cell growth. The identified upstream regulators have essential roles in signal transduction and cellular homeostasis. Furthermore, the overexpression of Hsp22 helped restore the levels of these proteins towards non-transgenic levels in rTg4510 mice. Taken together, this work demonstrates that Hsp22 overexpression affects global proteomic changes in neurons to protect against neurotoxic insults, like tau. This knowledge can serve as a foundation to enable future research on the characterization of novel functions of chaperones and their ability to maintain neuronal health and function.

## 4. Materials and Methods

### 4.1. Virus Production

wtHsp22, mtHsp22, and GFP were subcloned into pTR12.1-MCS vector containing a short hybrid CMV/chicken β-actin promoter [67]. Plasmids were confirmed by sequencing. wtHsp22-pTR12.1, mtHsp22-pTR12.1, or GFP-pTR12.1 were co-transfected with helper plasmids pFΔ6 and pAAV9 into HEK293T using polyethyleneimine to generate AAV9 particles. After the recombinant virus was harvested by three cycles of freeze–thaw, the crude lysate was clarified by centrifugation and purified using four iodixanol gradients. After concentration, a SYBR-green-based real-time PCR was used to determine the viral titer in genomes/mL [68].

### 4.2. Animals

Procedures involving animal subjects were approved and conducted following the guidelines set by the University of South Florida Institutional Animal Care and Use Committee. The rTg4510 and parental mice were maintained and genotyped as described previously [38]. Surgical procedures were performed as described previously [69]. Briefly, following analgesic, mice were anesthetized with isoflurane. Bilateral injections of 2 µL of AAV9 at a concentration greater than 10^12^ vg/mL were performed using a robotic stereotaxic apparatus (Neurostar) in the hippocampus: X= ±3.6 mm, Y= −3.5 mm, and Z = +2.68 from bregma and frontal cortex: −1.5 mm anteroposterior, ±2 mm bilateral, and +3.0 mm vertical from bregma at a rate of 5 µL/minute using convection-enhanced delivery (CED) [70]. After 2 min, the needle was removed from site, and the incision was closed with wound clips. All mice were blinded and used for behavioral testing, and then the mice were randomly assigned to two groups for evaluation by electrophysiology or immunohistochemistry and MS/immunoblotting, which is summarized in Table 1.

### 4.3. Radial-Arm Water Maze with Reversal and Open Field

To evaluate spatial learning and memory, at 6 months of age, non-transgenic (*n* = 31; N_wtHsp22_ = 11 (6M/5F), N_mtHsp22_ = 10 (5M/5F), N_GFP_ = 10 (5M/5F)) and rTg4510 (*n* = 30; N_wtHsp22_ = 10 (5M/5F), N_mtHsp22_ = 10 (5M/5F), N_GFP_ = 10 (5M/5F)) mice were evaluated by hippocampal-dependent two-day RAWM with reversal [71]. In this task, mice are trained over the course of three days using spatial cues to locate a hidden escape platform in a 6-arm maze submerged in water. On day one, 12 trials were run in four blocks of three. After each three-trial block, a second group of mice was run, allowing for a rest period before mice were exposed to the second block of three trials. The goal arm was different for each mouse in order to minimize odor cues. The start arm was varied based on a random number generator for each trial, while keeping the platform arm constant for individual mice on day one and two. For the first day, the platform was hidden every other trial. On day two, the platform remained hidden. Day three, the platform was moved to the opposite goal arm (reversal) and remained hidden. Number of errors (incorrect arm entries) was measured in a one-minute time frame. An error was also added if no arm choice was made within 20 s. Blind numbers were assigned by a separate individual to each mouse to eliminate biases, and to minimize individual-mouse trial variability, errors for three consecutive trials were averaged for each mouse, producing four data points for each day. Statistical analysis was determined by two-way ANOVA and a Bonferroni multiple comparisons test.

For open-field testing, 6-month-old non-transgenic (*n* = 31; N_wtHsp22_ = 11 (6M/5F), N_mtHsp22_ = 10 (5M/5F), N_GFP_ = 10 (5M/5F)) and rTg4510 (*n* = 30; N_wtHsp22_ = 10 (5M/5F), N_mtHsp22_ = 10 (5M/5F), N_GFP_ = 10 (5M/5F)) mice were monitored for 10 min in an open field (Stoelting). Activity, regarding total distance and time spent in the center, was recorded and analyzed using ANY-maze video tracking software version 5.11 (Stoelting, Wood Dale, IL, USA). Statistical analysis was determined by two-way ANOVA and Bonferroni multiple comparisons test.

### 4.4. Immunohistochemistry, Immunoblotting, and Gallyas Staining

For immunohistochemistry, rTg4510 (*n* = 6/AAV; 3M and 3F) mice were harvested, and tissues were processed as described previously [72]. Briefly, mice were perfused with saline following pentobarbital-solution overdose. Brains were extracted and cut along the mid-line. One hemisphere was submerged in 4% paraformaldehyde for immunohistochemistry, and the other hemisphere was snap-frozen for mass spectrometry, described below. Fixed tissues were cryoprotected and sectioned horizontally (25 μm, with a 50 µm section every 8th section) on a sliding microtome with freezing stage. For immunoblotting, rTg4510 (*n* = 9, N_wtHsp22_ = 3 (2M/1F), N_mtHsp22_ = 3 (1M/2F), and N_GFP_ = 3 (2M/1F)). For immunoblotting, hippocampal brain-tissue lysates were prepared through resuspension in 10µL/mg of ABC buffer (50 mM ammonium bicarbonate, pH 7.55, 5% SDS, with protease inhibitor cocktail EDTA-free, phenylmethylsulfonyl fluoride, protease inhibitor 2, and protease inhibitor 3, all with a 1:100 dilution) and sonicated 2 times (20% amplitude, 5 s on, 5 s off for 15 s) while on ice. Samples were spun at 13,000× *g* for 15 min at 4 °C. Supernatant from each tube was collected, and protein concentration was measured using BCA protein assay kit from ThermoFisher (Waltham, MA, USA) for Western blot analysis for the following antibodies: rabbit anti-human tau (1:1000; Agilent, Santa Clara, CA, USA; A002401-2), rabbit anti-pS262 tau (1:1000; AnaSpec, Fremont, CA, USA; 54973), rabbit anti-pT231(1:1000; AnaSpec; 55313), rabbit anti-EIF4E (1:1000; Cell Signaling; 9742), mouse anti-IκBα (1:1000; Cell Signaling, Danvers, MA, USA; 4814), rabbit anti-T22 (1:1000; Millipore, Burlington, MA, USA; ABN454), rabbit anti- β-Tubulin (1:1000; Cell Signaling; 2128), rabbit anti-GAPDH (1:1000; Proteintech, Rosermont, IL, USA; 10494-1), and mouse anti-GAPDH (1:1000; Proteintech; 60004-1). Immunostaining was performed using the following antibodies: rabbit anti-Hsp22 (1:300; Stress Marq, Victoria, BC, Canada; SPC181D), rabbit anti-human tau (1:100,000; Agilent; A002401-2), rabbit anti-pS262 tau (1:1000; AnaSpec; 54973), rabbit anti-pT231(1:300; AnaSpec; 55313), rabbit anti-p62 (1:300; Cell Signaling; 5114S), rabbit anti-T22 (1:15,000; Millipore; ABN454), rabbit anti-doublecortin (DCX; 1:1000; Cell Signaling; 4604S) and developed using 1.4 mM diaminobenzidine and nickel with 0.03% H_2_O_2_ following incubation with biotinylated goat anti-rabbit secondary antibody (1:3000; Southern Biotech, Birmingham AL, USA; 4050-08) and Vectastain ABC kit (Vector Laboratories, Burlingame, CA, USA; PK-4000). We also looked for GFP fluorescence to confirm viral expression. Gallyas silver-staining method was performed as previously described [24,72,73]. Stained slides were imaged using a 20× objective on a Zeiss AxioScan.Z1 (ZEISS Microscopy, White Plains, NY, USA) slide scanner. NearCYTE (http://nearcyte.org) was used to perform densiometric analyses in the hippocampus. Regions were drawn based on landmarks in the mouse brain atlas by a blinded experimenter. Statistical analysis was determined by one-way ANOVA and Tukey’s multiple comparison test for Dako, pS262, pT231, Gallyas silver stain, Dunnett’s for T22, DCX, and p62, and *t*-test for viral expression validation.

### 4.5. Unbiased Stereology

To evaluate effects on neuronal health, unbiased stereology was performed, using an optical fractionator method of stereological counting with stereological software (Stereologer 2000 Version 3.0 CP—Version Two; Stereology Resource, Tampa, FL, USA). Sections were stained with biotin-conjugated NeuN (1:3 K; Millipore; MAB377BMI), as described above, but without nickel development. Tissues were mounted and counterstained by cresyl violet and cover-slipped. Slides were analyzed by a blinded investigator. A systematic random sampling of sections was coded within the DG, which is defined at 4× magnification based on a mouse brain atlas to maintain consistency. Then, a computer-generated grid was placed randomly over the area of interest at 100× magnification, and cells were counted within three-dimensional optical dissectors if they were within the dissector or touching the green lines and were excluded if they were outside the dissector or touching the red lines. Section thickness was measured in all sections to estimate mean section thickness for each animal after tissue processing. Statistical analysis was determined by one-way ANOVA and Dunnett’s multiple comparisons test.

### 4.6. Electrophysiology

A subset of the AAV-injected mice, non-transgenic (*n* = 12; N_wtHsp22_ = 10 slices form 4 (2M/2F), N_mtHsp22_ = 31 slices from 4 (2M/2F), N_GFP_ = 31 slices from 4 (2M/2F)) and rTg4510 mice (*n* = 12; N_wtHsp22_ = 12 slices from 4 (2M/2F), N_mtHsp22_ = 18 slices from 4 (2M/2F), N_GFP_ = 13 slices from 4 (2M/2F)) was evaluated for electrophysiological changes by a blinded investigator using LTP, as previously described [74,75]. Hippocampal slices (400 µm) were perfused in artificial CSF (ACSF) at 1 mL/minute, and fEPSPs were obtained from Schaffer collaterals using a 0.1 ms biphasic pulse every 20 s, a bipolar Teflon-coated platinum stimulating electrode, and a glass recording microelectrode filled with ACSF (resistance of 1–4 mΩ). After a consistent pulse was determined for a 5–10 min period, threshold voltage for evoking a fEPSP was established, and the voltage was raised 0.5 mV until maximum amplitude of the fEPSP was reached, generating the I/O curve. The fEPSP baseline response was defined at 50% of the stimulus voltage used to produce maximum fEPSP response (amplitude) in the I/O curve. Baseline was recorded for 20 min, and tetanus Theta-burst stimulation (5 trains of 4 pulse bursts at 200 Hz separated by 200 ms, repeated 6 times with an intertrain interval of 10 s) was then used to induce LTP, which was recorded for 60 min. The potentiation signal was then normalized to the mean baseline fEPSP descending slope for each tissue slice to generate the data. Data were removed if baselines values exceeded 20% variability. Statistical analysis of the last five minutes of LTP was determined by one-way ANOVA and Dunnett’s post hoc test.

### 4.7. Mass Spectrometry

Hippocampal brain lysates from all injected mice were prepared for suspension-trap (S-trap) proteomics. Tissue was lysed in 10µL/mg of ABC buffer (50 mM ammonium bicarbonate, pH = 7.55, 5% SDS, with protease inhibitor cocktail EDTA free, phenylmethylsulfonyl fluoride, protease inhibitor 2, and protease inhibitor 3, all with a 1:100 dilution) and sonicated 2 times (20% amplitude, 5 s on, 5 s off for 15 s) while on ice. Samples were spun at 13,000× *g* for 15 min at 4 °C and reduced, with a final concentration of 20 mM DTT, heated at 95 °C for 10 min, then cooled before alkylating cysteines with the addition of 40 mM iodoacetamide, final concentration. Samples were incubated in the dark for 20 min at room temperature, followed by removal of undissolved matter by centrifugation at 17,000× *g* for 10 min. The clear supernatant was transferred to a new tube, followed by the addition of 12% aqueous phosphoric acid at 1:10 dilution for a final concentration of 1.2% phosphoric acid. A total of 6 times the volume of S-trap binding buffer (90% aqueous methanol, 100 mM Tris, pH = 7.1) was added to acidified protein and then mixed well. The S-trap microcolumn was placed in a 1.7 mL tube in order to retain flowthrough. The sample mixture was added to the micro column 200 µL at a time, followed by centrifugation of the micro column 4000× *g* for 2 min. Then, flowthrough was removed when needed, and this process was repeated until all sample had gone through the column. The protein bound within the column was washed four times with 150 µL of S-trap buffer, with centrifugation and removal of flowthrough each time. S-trap was moved to a clean 1.7 mL sample tube for proteolytic digestion, where 20 µL of digestion buffer containing 50 mM ABC buffer with 1 ug Trypsin/Lys-C protease (Promega) was added to the microcolumn (to ensure no air bubbles remained between the protease digestion solution and the protein trap, gel-loading tips were used). The S-trap micro column was capped to limit evaporation loss without forming an airtight seal and was incubated in a heat block for 1 h at 47 °C for trypsin digestion. After digestion, peptides were eluted, first with 40 µL of 50 mM ABC, and centrifuged at 4000× *g* for 2 min. An additional 40 µL of 0.2% formic acid in LC-MS-graded water was then added and centrifuged at 4000× *g* for 2 min. Finally, to recover hydrophobic peptides, a final elution of 35 µL of 50% acetonitrile containing 0.2% formic acid was added, with a final centrifugation at 5000× *g* for 5 min. All elutes were collected in the same tube in order to prevent transfer loss. Eluted peptides were centrifuged under vacuum until dry and then resuspend in 0.1% formic acid in water. Samples were sonicated 10 min in a water bath and centrifuged at 17,000× *g* for 30 min to remove any non-soluble particulates before transferring the clarified peptide supernatants into autosampler vials.

The samples were then assessed by liquid chromatography-mass spectrometry (LC-MS) analysis. Peptides were analyzed using a Thermo Q-Exactive HF-X mass spectrometer coupled to a Thermo Easy nLC 1200. Samples were loaded into a precolumn, Acclaim PEPMAP 100 (75 µM, 2 cm, c18 3 µm, 100 Å), and then the trapped peptides were eluted at 300 nL/minute into an Acclaim PEPMAP 100 analytical column (75 µm, 25 cm, c18, 100 A) using a 180 min gradient with an initial starting condition of 2% buffer B (0.1% formic acid in 90% Acetonitrile) and 98% buffer A (0.1% formic acid in water). Buffer B was increased to 25% over 120 min, then raised to 35% after an additional 25 min. The gradient was increased up to 98% B over another 25 min, and high B (98%) was run for 10 min afterwards. The MS was outfitted with a Thermo Nanospray Flex Source with the following parameters: spray voltage: 2.24, capillary temperature: 300 dC, funnel RF level = 40. Parameters for data acquisition were as follows: For MS data, the resolution was 60,000, with an AGC target of 3 × 10^6^ and a max IT time of 50 ms; the range was set to sequentially search ranges of 375–600 m/z, 600–800 m/z, and 800/1200 m/z, selecting up to 20 peaks to further investigate with MS/MS; MS/MS data were acquired with a resolution of 15,000, an AGC of 1e5, max IT of 25 ms, with an isolation window of 1.6 m/z and a dynamic execution of 25 s.

The resulting data were searched using Thermo Proteome Discoverer 2.2 software. A fully reviewed mouse database was downloaded from UniProt, which was used in the SEQUEST HT search, and a custom database was created using the protein sequences for the viruses used. A full trypsin digestion with a maximum of 2 missed cleavages was selected, including a precursor mass tolerance of 10 ppm and a fragment mass tolerance of 0.02 Da. Included modifications are oxidation, N-terminal acetylation, and carbamidomethylation. The resulting peptides were filtered for high confidence and validated with a confidence threshold of 0.01 (Target FDR). Data were searched using Max Quant (version 1.6.10.43). Samples were searched again using a fully reviewed mouse database from UniProt, combined with common contaminates, and concatenated with the reversed version of all sequences using the Andromeda search engine integrated into Max Quant, with trypsin as the enzyme, a maximum of two missed cleavages, and a minimum of 6 amino acids. FDR threshold was set to 0.01, and the “align across runs” feature was selected, with a mass error of 10 ppm and a retention time window of 4 min. The label-free quant feature was selected for all searches, with a minimum ratio of 1, and the resulting data were exported for further analysis.

#### 4.7.1. Experimental Design and Statistical Rationale

Sample size for MS: brain lysates from rTg4510 and non-transgenic mice injected with GFP (*n* = 6/genotype), wtHsp22 (*n* = 6 for rTg4510 and *n* = 7 for non-transgenic), or mtHsp22 (*n* = 6 for rTg4510 and *n* = 6 for non-transgenic). The reason we used this sample size was because the rest of the brain tissue was used for IHC. Two controls were used: a non-transgenic genotype control and a GFP treatment control. We quantified using LFQ (label-free quantification). Proteomic analysis identified 3918 proteins across all experimental groups (non/GFP, non/wtHsp22, non/mtHsp22, rTg4510/GFP, rTg4510/wtHsp22, and rTg4510/mtHsp22). We then examined for differentially expressed proteins between groups and filtered using z-score cutoff (|z-score| > 1) to provide a more stringent dataset with a lower false-discovery rate (FDR). We detected 638 proteins in the *non/wtHsp22* vs. *non/GFP* comparison, 577 proteins in the *non/mtHsp22* vs. *non/GFP* comparison, 690 proteins in the *non/wtHsp22* vs. *non/mtHsp22* comparison, 582 proteins in the *rTg4510/wtHsp22* vs. *rTg4510/GFP* comparison, 569 proteins in the *rTg4510/mtHsp22* vs. *rTg4510/GFP* comparison, 652 proteins in the *rTg4510/wtHsp22* vs. *rTg4510/mtHsp22* comparison, and 747 proteins in the *rTg4510/GFP* vs. *non/GFP* comparison.

#### 4.7.2. Parallel Reaction Monitoring

A parallel reaction monitoring (PRM) program was set up using the Skyline version 20.2 software from the MacCoss lab downloadable program website, which calculated both the appropriate collision energy and ion masses for each peptide [76,77]. Peptides were selected from the previous data-dependent analysis to include one from each protein of interest and one from GAPDH to act as a normalization factor. Samples were then rerun on the Q-exactive, with a scheduled program to increase sensitivity and reduce interference. Instrument methods were identical to the previously described run, except that the MS/MS acquisition time was increased to 150 ms. The resulting data were then analyzed again with the Skyline software, and the total peak area was exported for each peptide and sample by *t*-test towards GFP-injected mice.

### 4.8. Statistical Analysis

Statistical analysis was done through *t*-test, one-way ANOVA, two-way ANOVA, and respective post hoc test, as described in the legends. Graphs and statistical analyses were plotted and completed using GraphPad 9 (Prism).

## Figures and Tables

**Figure 1 ijms-23-00851-f001:**
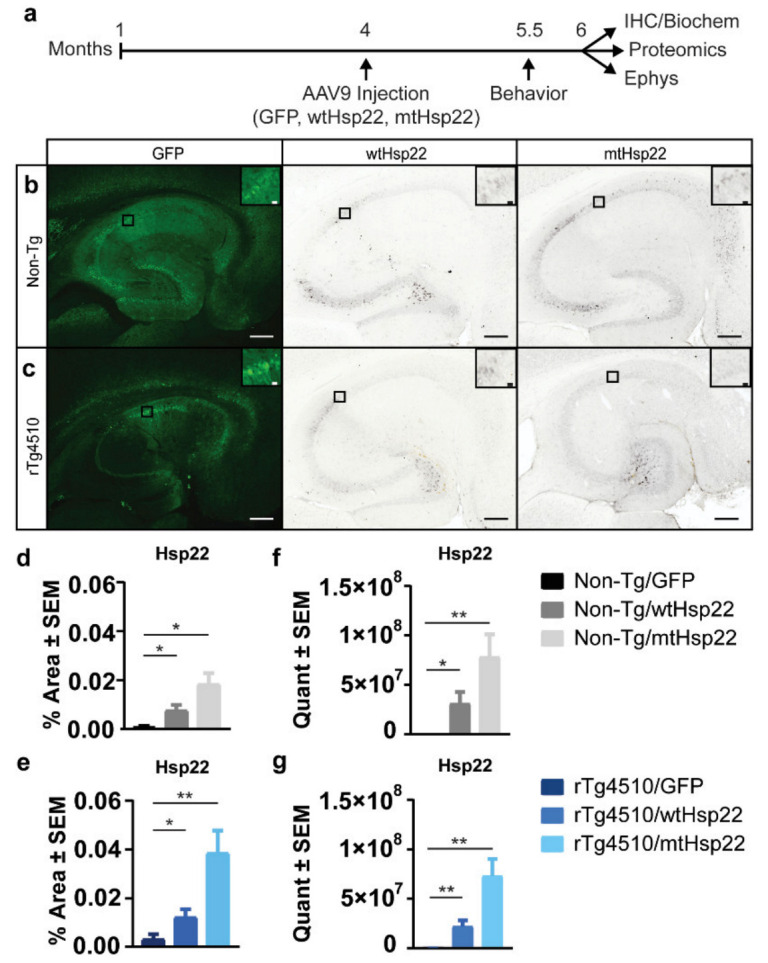
Hsp22 levels were significantly upregulated by AAV9. (**a**) Timeline of in vivo study. (**b**) Representative images of GFP, wtHsp22, and mtHsp22 levels in non-transgenic (Non-Tg) mice. (**c**) Representative images of GFP, wtHsp22, and mtHsp22 levels in rTg4510 mice. Scale bar represents 200 µm; inset represents 20 µm. (**d**) Quantification of immunohistochemical staining of Hsp22 levels in non-Tg mice (mean ± SEM, * *p* < 0.05) and (**e**) rTg4510 mice (mean ± SEM, * *p* < 0.05, ** *p* < 0.01) GFP (*n* = 5/Non-Tg; *n* = 7/rTg4510), wtHsp22 (*n* = 7/Non-Tg; *n* = 6/rTg4510) mtHsp22 (*n* = 6/Non-Tg; *n* = 6/rTg4510). (**f**) Quantification of Hsp22 levels in non-Tg mice by parallel reaction monitoring (mean ± SEM, * *p* < 0.05, ** *p* < 0.01). (**g**) Quantification of Hsp22 levels in rTg4510 mice by quantitative mass spectrometry parallel reaction monitoring (mean ± SEM, ** *p* < 0.05). GFP (*n* = 6/Non-Tg; *n* = 6/rTg4510), wtHsp22 (*n* = 7/Non-Tg; *n* = 6/rTg4510) mtHsp22 (*n* = 6/Non-Tg; *n* = 6/rTg4510).

**Figure 2 ijms-23-00851-f002:**
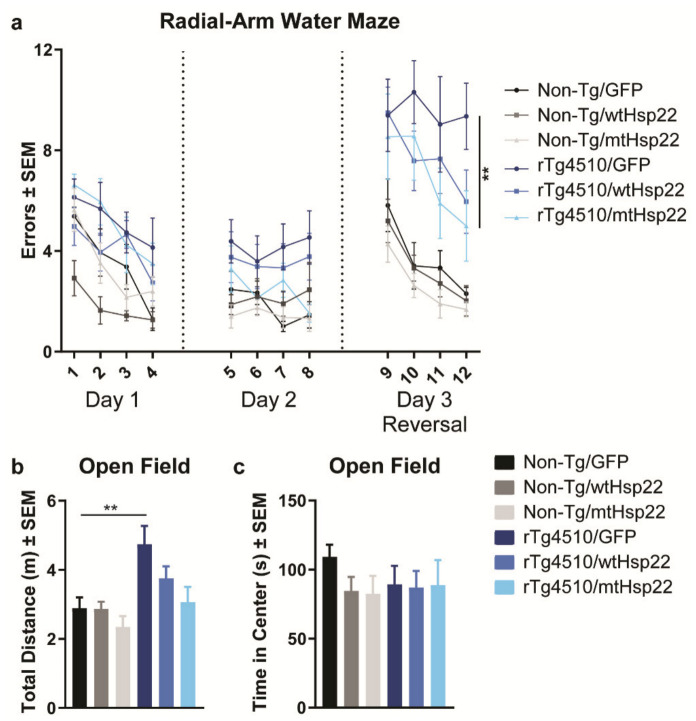
mtHsp22 overexpression rescues spatial reversal learning and memory in rTg4510 mice. (**a**) The 2-day Radial-Arm Water Maze task with reversal was used to assess spatial memory in non-transgenic (Non-Tg) and rTg4510 mice injected with GFP (*n* = 10/Non-Tg; *n* = 10/rTg4510), wtHsp22 (*n* = 11/Non-Tg; *n* = 10/rTg4510), or mtHsp22 (*n* = 10/Non-Tg; *n* = 10/rTg4510). Mice were tested in 12 trials per day for training (Day 1), testing (Day 2), and reversal (Day 3), and the number of errors was recorded. Average errors over each set of three trials is shown (mean ± SEM, ** *p* < 0.01). Non-Tg and rTg4510 mice injected with GFP (*n* = 10/Non-Tg; *n* = 10/rTg4510), wtHsp22 (*n* = 11/Non-Tg; *n* = 10/rTg4510), or mtHsp22 (*n* = 10/Non-Tg; *n* = 10/rTg451)0 were also evaluated in an open field for (**b**) total distance traveled in meters (m) and (**c**) time spent in the center in seconds (s) over the course of 10 min (mean ± SEM), ** *p* < 0.0027.

**Figure 3 ijms-23-00851-f003:**
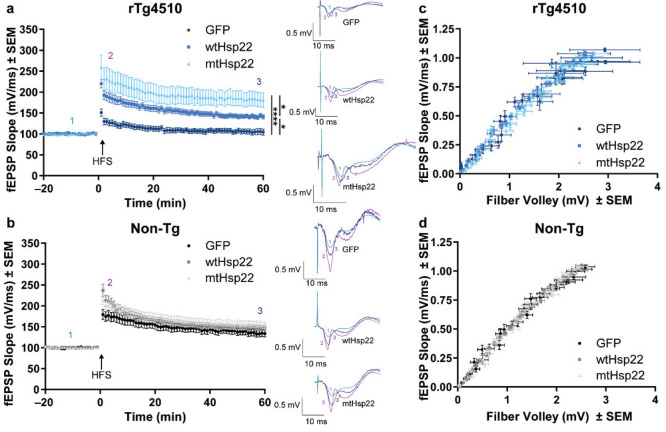
Hsp22 overexpression increases synaptic plasticity in rTg4510 mice. After recording a 20-min baseline, long-term potentiation (LTP) was induced in the Schaffer collaterals with high-frequency stimulation (HFS) (5 bursts of 200 Hz separated by 200 ms, repeated 6 times with an intertrain interval of 10 s). fEPSP was recorded for 60 min in (**a**) rTg4510, and (**b**) non-transgenic (Non-Tg) mice injected with GFP (*n* = 4/Non-Tg; *n* = 4/rTg4510), wtHsp22 (*n* = 4/Non-Tg; *n* = 4/rTg4510), or mtHsp22 (*n* = 4/Non-Tg; *n* = 4/rTg4510) (mean ± SEM, * *p* < 0.05, **** *p* < 0.0001). Representative traces are shown: 1 (teal) indicates baseline, 2 (pink) indicates early LTP potentiation in the first 3 min following HFS, and 3 (blue) indicates late LTP in the last 3 min of recording. The input/output curves of the fEPSP slope (mV/ms) versus the fiber volley amplitude (mV) in (**c**) rTg4510 and (**d**) Non-Tg mice injected with GFP, wtHsp22, or mtHsp22 (mean ± SEM).

**Figure 4 ijms-23-00851-f004:**
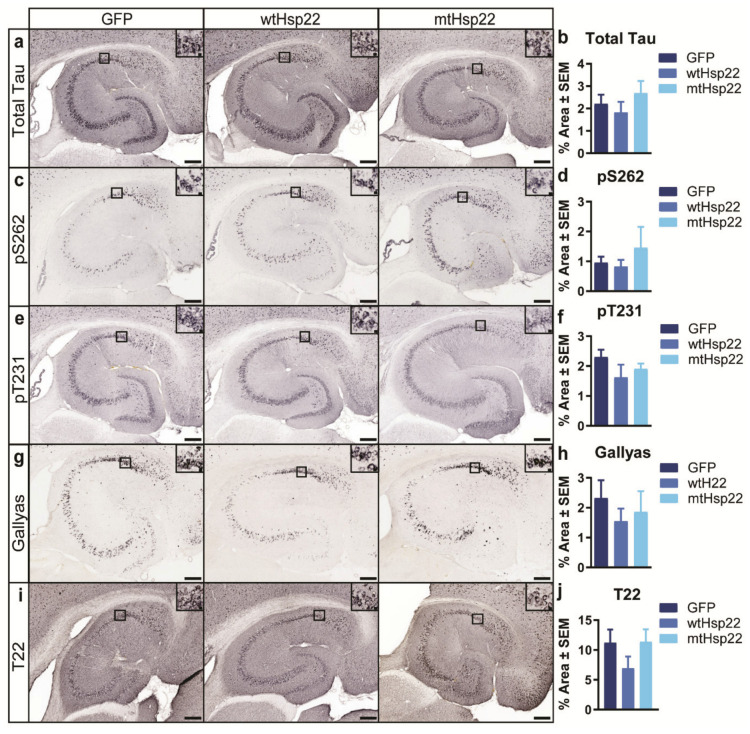
Overexpression of wtHsp22 and mtHsp22 does not alter tau accumulation. Immunohistochemical staining and quantification of the hippocampi of rTg4510 mice injected with GFP (*n* = 6), wtHsp22 (*n* = 6), or mtHsp22 (*n* = 6) for (**a**,**b**) total tau; (**c**,**d**) pS262 tau; (**e**,**f**) pT231 tau; (**g**,**h**) Gallyas silver, and (**i**,**j**) T22 oligomeric tau (mean ± SEM; scale bar represents 200 µm; inset represents 10 µm).

**Figure 5 ijms-23-00851-f005:**
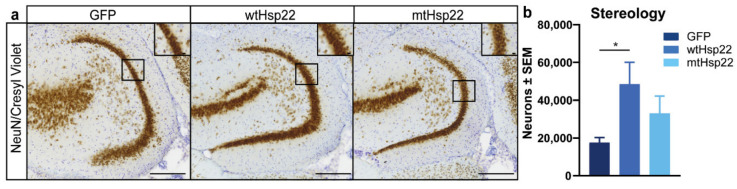
Hsp22 overexpression protects neurons in rTg4510 mice. (**a**) Immunohistochemistry staining and (**b**) quantification of neurons (NeuN/Cresyl violet) in the hippocampi of rTg4510 mice injected with GFP (*n* = 6), wtHsp22 (*n* = 6), or mtHsp22 (*n* = 6) (mean ± SEM; scale bar represents 200 µm; inset represents 10 µm). * *p* < 0.05.

**Figure 6 ijms-23-00851-f006:**
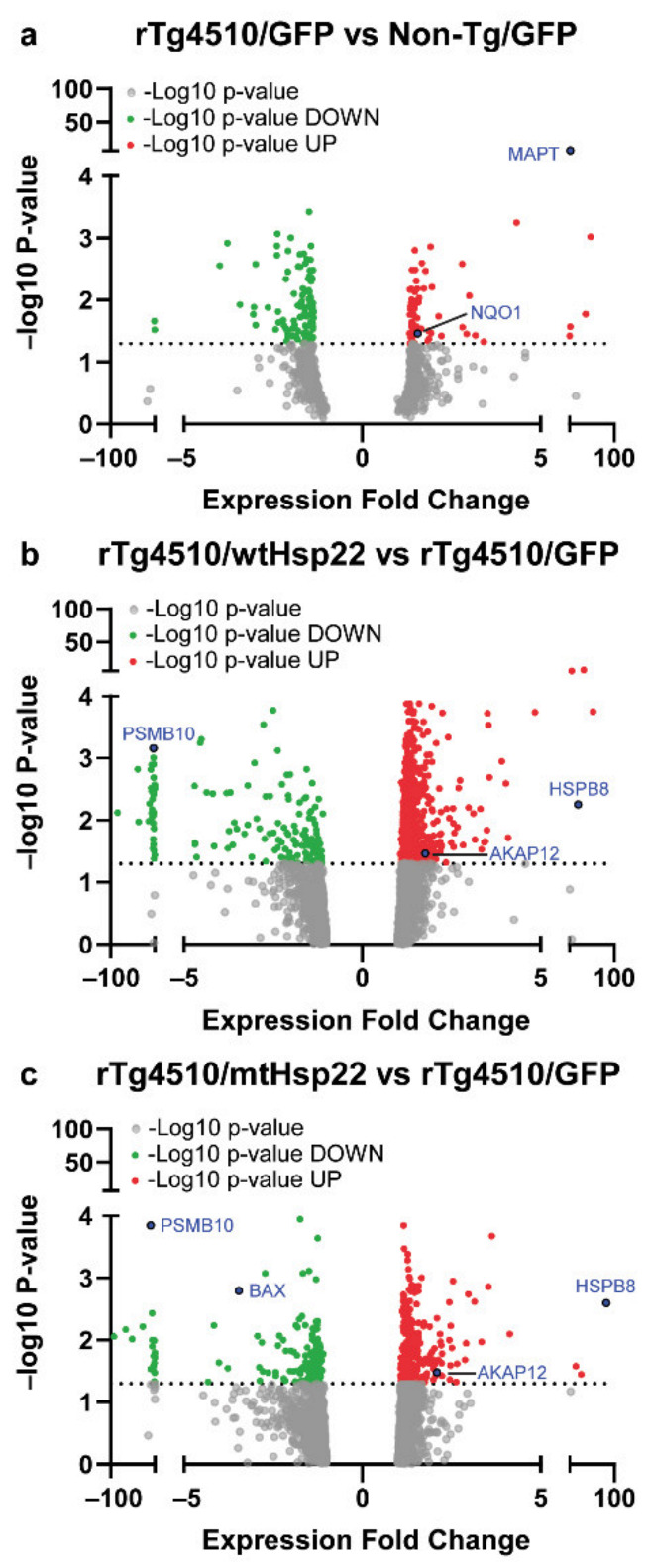
wtHsp22 and mtHsp22 overexpression significantly regulates key neuroprotective proteins. Protein-fold change was calculated using average LFQ intensities among samples where rTg4510/GFP was compared to non-transgenic (Non-Tg)/GFP, rTg4510/mtHsp22 was compared to rTg4510/GFP, and rTg4510/wtHsp22 was compared to rTg4510/GFP. (**a**) *rTg4510/GFP* vs. non-Tg*/GFP* ($\frac{rTg4510\;GFP}{\mathrm{Non}-\mathrm{Tg}\;GFP})$; (**b**) *rTg4510/mtHsp22* vs. *rTg4510/GFP* ($\frac{rTg4510\;mtHsp22}{rTg4510\;GFP})$; (**c**) *rTg4510/wtHsp22* vs. *rTg4510/GFP* ($\frac{rTg4510\;wtHsp22}{rTg4510\;GFP})$. Expression-fold change values were plotted against the corresponding –log_10_ (*p*-value) for each protein to generate volcano plots; horizontal dotted line represents the cutoff for statistical significance via Welch’s *t*-test (*p* > 0.05).

**Figure 7 ijms-23-00851-f007:**
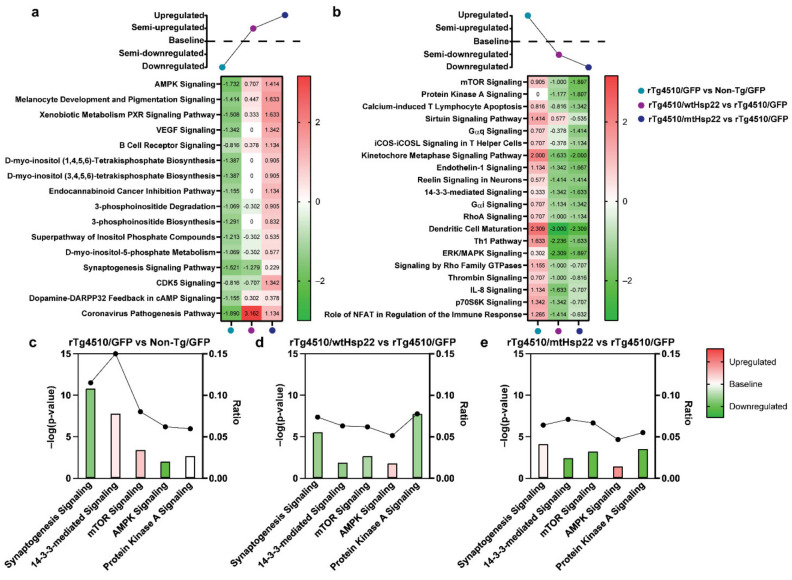
#wtHsp22 and mtHsp22 overexpression shifts key canonical pathways—AMPK signaling, synaptogenesis signaling, mTOR signaling, PKA signaling, and 14-3-3-mediated signaling in rTg4510 mice towards non-transgenic levels. Comparison analysis of canonical pathways annotated by IPA’s core analysis module regulated by *rTg4510/GFP* vs. *Non-transgenic (Non-Tg)/GFP*, *rTg4510/wtHsp22* vs. *rTg4510/GFP*, and *rTg4510/mtHsp22* vs. *rTg4510/GFP* comparison analysis. (**a**) Heat map based on trend and z-score, where *rTg4510/GFP* vs. *WT/GFP* had pathways that were downregulated based on genotype effect, *rTg4510/wtHsp22* vs. *rTg4510/GFP* had pathways that were semi-upregulated based on treatment, and *rTg4510/mtHsp22* vs. *rTg4510/GFP* had pathways that were upregulated based on treatment. (**b**) Heat map based on trend and z-score, where *rTg4510/GFP* vs. *Non-Tg/GFP* had pathways that were upregulated based on genotype effect, *rTg4510/wtHsp22* vs. *rTg4510/GFP* had pathways that were semi-downregulated based on treatment, and *rTg4510/mtHsp22* vs. *rTg4510/GFP* had pathways that were downregulated based on treatment. (**c**) Key canonical signaling pathways annotated by IPA’s core analysis module for *rTg4510/GFP* vs. *Non-Tg/GFP*; (**d**) *rTg4510/wtHsp22* vs. *rTg4510/GFP;* and (**e**) *rTg4510/mtHsp22* vs. *rTg4510/GFP*. X axis indicates pathways (synaptogenesis signaling, 14-3-3-mediated signaling, mTOR signaling, AMPK signaling, protein-kinase-A-signaling); left Y axis denotes -log(*p*-value) derived from Fischer’s exact test, right-tailed; right Y axis (points on black line) represents ratio of dataset proteins to total known proteins for that pathway. Red bars indicate predicted activation (positive z-score), green bars indicate predicted inhibition (negative z-score), and white bars represent no activation or inhibition (0 z-score).

**Figure 8 ijms-23-00851-f008:**
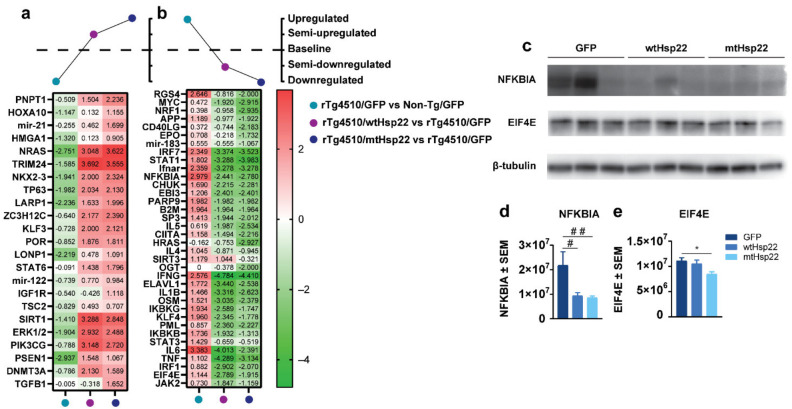
wtHsp22 and mtHsp22 overexpression shifts key upstream regulators EIF4E and NFKBIA towards non-transgenic levels in rTg4510 mice. Comparison analysis of upstream regulators annotated by IPA’s core analysis module regulated by *rTg4510/GFP* vs. *Non-transgenic (Non-Tg)/GFP*, *rTg4510/wtHsp22* vs. *rTg4510/GFP*, and *rTg4510/mtHsp22* vs. *rTg4510/AAV9 GFP*. (**a**) Heat map based on trend and z-score, where *rTg4510/GFP* vs. *Non-Tg/GFP* had upstream regulators that were downregulated based on genotype effect, *rTg4510/wtHsp22* vs. *rTg4510/GFP* had upstream regulators that were semi-upregulated based on treatment, and *rTg4510/mtHsp22* vs. *rTg4510/GFP* had upstream regulators that were upregulated based on treatment. (**b**) Heat map based on trend and z-score, where *rTg4510/GFP* vs. *Non-Tg/GFP* had upstream regulators that were upregulated based on genotype effect, *rTg4510/wtHsp22* vs. *rTg4510/GFP* had upstream regulators that were semi-downregulated based on treatment, and *rTg4510/mtHsp22* vs. *rTg4510/GFP* had upstream regulators that were downregulated based on treatment. Immunoblotting and quantification of hippocampi of rTg4510 mice with GFP (*n* = 3), wtHsp22 (*n* = 3), or mtHsp22 (*n* = 3). (**c**) Immunoblot of EIF4E and NFKBIA. (**d**,**e**) Quantification of EIF4E and NFKBIA (mean ± SEM, # *p* < 0.07, ## *p* < 0.06, * *p* < 0.04).

**Table 1 ijms-23-00851-t001:** Summary of mice used for AAV9 study.

Experiment	Non-Tg	rTg4510
Radial-Arm Water Maze	*n* = 31; wtHsp22 = 11 (6M/5F), mtHsp22 = 10 (5M/5F), GFP = 10 (5M/5F)	*n* = 30; wtHsp22 = 10 (5M/5F), mtHsp22 = 10 (5M/5F), GFP = 10 (5M/5F)
Open-Field Testing	*n* = 31; wtHsp22 = 11 (6M/5F), mtHsp22 = 10 (5M/5F), GFP = 10 (5M/5F)	*n* = 30; wtHsp22 = 10 (5M/5F), mtHsp22 = 10 (5M/5F), GFP = 10 (5M/5F)
Immunohistochemistry	N/A	*n* = 18; wtHsp22 = 6 (3M/3F), mtHsp22 = 6 (3M/3F), GFP = 6 (3M/3F)
Immunoblotting	N/A	*n* = 9; wtHsp22 = 3 (2M/1F), mtHsp22 = 3 (1M/2F), GFP = 3 (2M/1F)
Electrophysiology	*n* = 12; wtHsp22 = 10 slices from 4 (2M/2F), mtHsp22 = 31 slices from 4 (2M/2F), GFP = 31 slices from 4 (2M/2F)	*n* = 12; wtHsp22 = 12 slices from 4 (2M/2F), mtHsp22 = 18 slices from 4 (2M/2F), GFP = 13 slices from 4 (2M/2F)
Mass Spectrometry	*n* = 19; wtHsp22 = 7 (4M/3F), mtHsp22 = 6 (3M/3F), GFP = 6 (3M/3F)	*n* = 18; wtHsp22 = 6 (3M/3F), mtHsp22 = 6 (3M/3F), GFP = 6 (3M/3F)

## Data Availability

The datasets used for Figure 1, Figure 2, Figure 3, Figure 4, Figure 5, Figure 6, Figure 7 and Figure 8 and Appendix A will be made available from the corresponding author on reasonable request. Data used for Appendix A can be found on the following public databases: https://www.uniprot.org/, http://d2p2.pro, https://string-db.org/.

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
