# Peer review of "Small Heat Shock Protein 22 Improves Cognition and Learning in the Tauopathic Brain"

_ijms, 2022, doi:10.3390/ijms23020851_

Round 1

Reviewer 1 Report

In this study Rodriguez-Ospina et al show that overexpression of wildtype Hsp22 and a phospho-mimetic mutant version protects neurons and improves synaptic plasticity, learning and memory capability in a mouse tauopathy model (rTg4510). Different from previous in vitro work the authors find that overexpression of Hsp22 in vivo does has no effect on the accumulation of tau. Instead, their proteomic analysis shows that wt and mutant Hsp22 overexpression restored pathways involved in cell growth, neuronal protection and cell metabolism. They validated the expression levels of EIF4E and NFKBIA, which are often upregulated in the brains of AD patients and downregulated upon overexpression of Hsp22, correlating with neuroprotection. Overall, this is an interesting and important study demonstrating that the beneficial effects of Hsp22 expression in vivo occur in a more complex interplay of cellular response pathways that tie with cellular metabolism and growth pathways. This study will need follow-up on how Hsp22 and its phospho-mimetic mutant induces the observed changes in potential cellular pathways affected by Hsp22 overexpression, but it nevertheless provides a solid basis for such work that will be important for other researchers in the field.

I only have minor comments to increase clarity.

Figure 1:

Immunohistochemical staining of Hsp22 and quantification is very difficult to judge. For example, in 1c the staining looks similar between wtHsp22 and mtHsp22, but the quantification in 1f shows that levels are different. I would suggest to either show a western blot if possible or emphasize the use of PRM being a method based on quantitative Mass-Spec that allows measuring Hsp22 levels more accurately. Referring to the results in Figure 6, where Mass-Spec is used and the same results on Hsp22 levels are validated will also be worthwhile.

Figure 8:

The authors only selected ELF4E and NFKBIA to validate their downregulated expression levels by Western Blot. Why were only these two proteins validated and proteins such as e.g. TRIM24 or SIRT1 that are upregulated upon Hsp22 overexpression dismissed for validation?

Line 69: Both references mentioned here (12, 23) do not seem to be correct.

Line 242: a reference to support this statement is needed here.

Author Response

Reviewer 1

Figure 1: Immunohistochemical staining of Hsp22 and quantification is very difficult to judge. For example, in 1c the staining looks similar between wtHsp22 and mtHsp22, but the quantification in 1f shows that levels are different. I would suggest to either show a western blot if possible or emphasize the use of PRM being a method based on quantitative Mass-Spec that allows measuring Hsp22 levels more accurately. Referring to the results in Figure 6, where Mass-Spec is used and the same results on Hsp22 levels are validated will also be worthwhile.

Thank you for your suggestions. We have selected a new representative image for 1c, based on your suggestion that better emphasizes the differences quantified. We have also better emphasized the mass spec PRM as a quantitative measure in the results section for Figure 1 and again in Figure 6. These changes can be found on Lines 84-85, Figure 1 legend, and Lines 159-160.

Figure 8: The authors only selected ELF4E and NFKBIA to validate their downregulated expression levels by Western Blot. Why were only these two proteins validated and proteins such as e.g. TRIM24 or SIRT1 that are upregulated upon Hsp22 overexpression dismissed for validation?

We did have TRIM24 and SIRT1 on our initial list to follow-up on. We used the human brain atlas as a first reference and removed proteins with minimal or no expression in brain tissues, which removed TRIM24 from our list. We have probed for SIRT1 with two antibodies but have not had success in getting positive bands from these antibodies. We carefully selected a new SIRT1 antibody, which was supposed to arrive mid-November, but has still not arrived and has no expected shipping date currently. We agree that this is an exciting target, and we expect to explore it further in the future. We appreciate your understanding of this limitation.

 Line 69: Both references mentioned here (12, 23) do not seem to be correct.

Thank you for pointing out this issue in the references. We have corrected these on Line 69 and have gone through the references one by one to correct any others that were not linked correctly.

Line 242: a reference to support this statement is needed here.

We have now added a reference to this statement on (now) Line 235. We apologize for omitting this in the prior submission.

Reviewer 2 Report

Rodriguez Ospina reports interesting findings regarding probable role of Heat shock protein 22 (Hsp22) in improving cognition and learning in taupathic brain. Overall, this is a well-written manuscript— statistical analysis, data reporting, analysis and interpretation of results appear sound. Figures and images are high quality and were described clearly. This reviewer suggest these few minor modifications to the paper to further improve its clarity.

Tissue processing took place after behavioral tests. What was the criteria for selecting animals for subsequent immunohistochemistry/immunoblotting? This needs to be stated in the manuscript.

Some tests were conducted on both male and female mice. Groups include both male and female subjects. Were there tests used to investigate gender difference in some outcomes/parameters (e.g. behavioral tests).  Moreover, were representative histology images taken from male or female mice?

SIRT1 has also been implication in AD. Were there attempts to validate expression level change in animal models?

The last 5 sentences in the Introduction could just be deleted as they pertain to “results” and implications of findings.

Abbreviations (NFKBIA, EIF4E, etc.) need to be written out in the Discussion section.

For consistency, the terms “in vivo” and “in vitro” should be italicized.

For consistency, number of animals per group (see Fig, 4) should be stated in Figure legends.

This sentence is confusing: “In this study, we found that tau transgenic mice overexpressing mtHsp22 or 218 wtHsp22, albeit to a lesser degree, improved cognition and synaptic plasticity”. Please revise to state that overexpression improved cognition and synaptic plasticity, or mice overexpressing Hsp22 showed improvement in cognition and so on.

Author Response

Thank you for your helpful comments. Please see the response in the attached document.
